# *CALICO*: Self-Supervised Camera-LiDAR Contrastive Pre-training for BEV Perception

**Jiachen Sun** [*1], **Haizhong Zheng** [1], **Qingzhao Zhang** [1], **Atul Prakash** [1], **Z. Morley Mao** [1], and **Chaowei Xiao** [2,3]

[1] University of Michigan
[2] University of Wisconsin, Madison
[3] NVIDIA

## Abstract

Perception is crucial in the realm of autonomous driving systems, where bird's eye view (BEV)-based architectures have recently reached state-of-the-art performance. The desirability of self-supervised representation learning stems from the expensive and laborious process of annotating 2D and 3D data. Although previous research has investigated pretraining methods for both LiDAR and camera-based 3D object detection, a unified pretraining framework for multimodal BEV perception is missing. In this study, we introduce *CALICO*, a novel framework that applies contrastive objectives to both LiDAR and camera backbones. Specifically, *CALICO* incorporates two stages: point-region contrast (PRC) and region-aware distillation (RAD). PRC better balances the region- and scene-level representation learning on the LiDAR modality and offers significant performance improvement compared to existing methods. RAD effectively achieves contrastive distillation on our self-trained teacher model. *CALICO*'s efficacy is substantiated by extensive evaluations on 3D object detection and BEV map segmentation tasks, where it delivers significant performance improvements. Notably, *CALICO* outperforms the baseline method by 10.5% and 8.6% on NDS and mAP. Moreover, *CALICO* boosts the robustness of multimodal 3D object detection against adversarial attacks and corruption. Additionally, our framework can be tailored to different backbones and heads, positioning it as a promising approach for multimodal BEV perception.

## 1 Introduction

The pursuit of on-road autonomous driving has sparked the study of various perception methods, necessitating a more in-depth understanding of the driving environment. A critical component in this landscape is bird's eye view (BEV) perception, an approach that presents a top-down 360° view, offering a comprehensive and intuitive understanding of the vehicle's surroundings. Pioneering works (Lang et al., 2019; Huang et al., 2021; Liu et al., 2023) have explored BEV perception for various modalities including LiDAR (Lang et al., 2019; Zhou & Tuzel, 2018), camera (Li et al., 2022e; Huang et al., 2021), and sensor fusion (Liu et al., 2023; Bai et al., 2022a). Despite the remarkable progress achieved in this domain, challenges still persist in optimizing the efficiency (Yin et al., 2022) and robustness (Sun et al., 2020a) of BEV perception systems. The development of these models has been predominantly reliant on massive labeled datasets, which are not only costly and laborious to acquire but also subject to inherent annotation biases (Chen & Joo, 2021).

Self-supervised learning (SSL) is a promising approach to harness the potential of unlabeled data (Jaiswal et al., 2020) that improves model training efficiency. It generally involves designing a pretext task in which supervision signals can be autonomously generated from the data itself, thus facilitating representation learning. When supplemented with a modest amount of labeled data for subsequent tasks, the learned representations can be finetuned, resulting in superior performance. The recent progress in SSL is largely attributable to the emergence of contrastive learning (He et al., 2020). Existing studies in this field have been devoted to classic 2D vision tasks (Chen et al., 2020b;a). The inter-instance discrimination pretext typically assumes objects of interest are centered in the

---

*jiachens@umich.edu

images, implying that global consistency suffices for effective representation learning. However, this assumption is not applicable in autonomous driving perception, which may include 10+ objects scattered in the BEV space per frame (Yin et al., 2022). Recent studies have delved into region-level contrastive learning, utilizing more granular intra-instance discrimination (Yin et al., 2022; Bai et al., 2022b; Xie et al., 2021; Wei et al., 2021) to enhance object detection representation learning. Yet, partitioning regions presents an inherent challenge in these methods as it requires to be label-free. Random assignments might obscure semantics for objects of different sizes (Wei et al., 2021; Xie et al., 2021), leading to suboptimal performance. Conversely, heavy reliance on heuristic assignments risks overfitting during finetuning (Bai et al., 2022b; Yin et al., 2022).

Besides unimodal contrastive pretraining, CLIP (Radford et al., 2021) first introduced contrastive objectives to multimodal feature embeddings, *i.e.,* images and texts, thereby enabling zero-shot recognition. The pretrained backbones are also demonstrated to be robust against various distribution shifts. GLIP (Li et al., 2022a) subsequently extended multimodal contrastive learning to object detection by grounding phrases in the textual input. While SimIPU (Li et al., 2022d) initially attempted contrastive pretraining across camera and LiDAR modalities in autonomous driving perception, its design is limited to the pixel space for the image input. We demonstrate it struggles to scale to prevalent BEV perception models due to the implicit pixel-to-BEV space transformation. Additionally, it solely concentrates on global invariant representation learning, neglecting object-level semantics. Therefore, devising a pretraining framework for multimodal BEV perception remains an open research problem.

In this paper, we propose a novel contrastive pretraining paradigm, *CALICO*, to address these challenges. This approach consists of two key components: point-region contrast (PRC) and region-aware distillation (RAD). Our preliminary analysis suggests that the region partitioning in previous studies inadequately captures object-level semantics. Consequently, we introduce a simple yet effective semantic pooling method for top-down region clustering to better encapsulate the object-aligned assignments. We further augment the pooled LiDAR points with their region assignment and consider the remaining ones as semantic-less points (§ 2.2). Our PRC adopts the finegrained point-wise operation on the LiDAR backbone to achieve both scene- and region-level contrast. Specifically, we first enhance the design in (Bai et al., 2022b) by employing semantic-less points as negative pairs to boost efficacy. Additionally, we introduce another loss term to balance scene- and region-level representation learning (§ 2.3). Once the LiDAR backbone is pretrained by PRC, we then apply RAD to the camera backbone. Our analysis reveals that the implicit transformation from the pixel to BEV space renders the initial embedding from the camera feature map meaningless. Hence, we propose to leverage contrastive distillation, *i.e.,* to stop gradient propagating to the LiDAR backbone, to train the camera backbone. We introduce a new objective that normalizes the weights of point-wise feature embeddings within the same region during distillation, which particularly is optimized for our self-supervised pretrained teacher models (Chen et al., 2022b) (§ 2.4).

Furthermore, we perform thorough evaluations of *CALICO* on 3D object detection and BEV map segmentation tasks using the nuScenes dataset (Caesar et al., 2020). The experimental results clearly show that our PRC achieves a significant 8.6% and 5.1% improvement on the LiDAR-only modality in terms of nuScene detection score (NDS) and mean average precision (mAP), respectively, compared to the baseline method, when fine-tuned on a small annotated subset. *CALICO* further extends this improvement to 10.5% and 8.6% on NDS and mAP, respectively. For the BEV map segmentation task, *CALICO* consistently surpasses the baseline methods by 5.7% in the mean intersection of union (mIoU) when finetuning on 5% of the labeled data. We also assess the robustness of models finetuned with our methods. We additionally leverage Waymo (Sun et al., 2020b) datasets to demonstrate the generalizability and transferability of our *CALICO*. Notably, *CALICO* enhances resistance against adversarial LiDAR spoofing attacks and distribution shifts by 45.3% and 12.8%, respectively. The ensuing sections of this paper will review several related topics, delve into more details of our methodology and extensive evaluation with ablation studies of our method with existing approaches, and highlight potential directions for future work in the field of autonomous driving perception.

## 2 METHODOLOGY

In this section, we comprehensively introduce our *CALICO* pretraining framework. In § 2.1, we briefly describe the existing designs and motivate the proposal of *CALICO*. We present the overview

of *CALICO* in § 2.2. Then, the two major components point-region contrast (PRC) and region-aware distillation (RAD) of *CALICO* are detailed in § 2.3 and § 2.4, respectively.

## 2.1 EXISTING DESIGNS AND MOTIVATION

Due to space limits, we put the thorough related work review in Appendix A and discuss the most relevant studies in this section. A few prior studies proposed contrastive pretraining frameworks for 3D object detection, including PointContrast (Xie et al., 2020), GCC-3D (Liang et al., 2021), ProposalContrast (Yin et al., 2022), and SimIPU (Li et al., 2022d), as introduced in § A. In this section, we aim to categorize these existing methodologies, providing a structured overview of their designs. This taxonomy will elucidate the motivation behind our proposal of *CALICO*, highlighting our unique attributes and improvements.

**Scene-Level Contrast**. PointContrast (Xie et al., 2020) is a pioneering study in enabling self-supervised pretraining for 3D point cloud architectures. Given the original point cloud $\mathcal{X} = \{\boldsymbol{x}_i\}_{i=0}^N$, PointContrast employs the InfoNCE loss (He et al., 2020) to contrastively learn point-wise features $\{\boldsymbol{z}_i^1\}_{i=0}^N$ and $\{\boldsymbol{z}_i^2\}_{i=0}^N$, which are extracted from two augmented views of the input point cloud, $\mathcal{X}^1$ and $\mathcal{X}^2$, respectively. Point-wise operation offers a finegrained and intuitive method for 3D point cloud learning, as it aligns seamlessly with the native representation of point clouds. However, PointContrast is fundamentally a scene-level contrast design, as it can be reduced to global invariant representation learning. This approach tends to overemphasize localization, which consequently leads to a limited perspective of objects of interest within a point cloud.

**Region-Level Contrast**. The concept of a region is crucial in object detection tasks. GCC-3D (Liang et al., 2021) employs temporal heuristics to extract moving points that cluster into meaningful regions. GCC-3D then applies RoIAlign (He et al., 2017) to the flattened 2D feature map, extracting region-level features for contrastive pretraining. This approach represents each region using a single, aggregated vector embedding for loss computation (Bai et al., 2022b). However, due to imperfect partitioning, object-level localization is often overlooked, as we have empirically demonstrated in Appendix B.

**Proposal-Aware Point Contrast**. ProposalContrast (Yin et al., 2022) aims to combine point-wise operation and region-level pretraining. It first randomly samples anchor points and then uses ball-queried neighbor points to aggregate "proposals" for anchors (Qi et al., 2017). A cross-attention module is applied to obtain proposal-aware features for each anchor point for contrastive pretraining. ProposalContrast achieves SOTA performance in the 3D object detection task on several benchmarks (Sun et al., 2020b; Mao et al., 2021). However, we discovered that ProposalContrast randomly samples anchor points and utilizes a fixed, deterministic radius for the ball query, resulting in proposals that lack object-level semantics.

**Camera-LiDAR Contrast**. In addition to pretraining on a single LiDAR modality, SimIPU (Li et al., 2022d) pioneers camera-LiDAR contrastive learning, which simultaneously applies point-level contrast pretraining to both intra and inter-modalities. We found that SimIPU underperforms in BEV perception architectures, as the transformation from pixel to BEV space is ***implicit***, causing the initialized features from the camera modality are insignificant. As the gradients computed by the contrastive loss will flow to both LiDAR and camera backbones, we demonstrate that SimIPU pretraining results in performance degradation in downstream multimodal BEV perception tasks.

## 2.2 OVERVIEW OF *CALICO*

To address the limitations of existing designs, we propose *CALICO*, which consists of two key stages: point-region contrast (RPC) and region-aware distillation (RAD), as shown in Figure 1. PRC achieves a more finegrained and balanced formulation of both point- and region-level contrastive pretraining on LiDAR point clouds. RAD is a dense distillation framework from LiDAR to camera feature maps, which takes into account the assigned region for each distilled point-level embedding.

Specifically, let $\mathcal{X}$ denote the pristine point cloud, where each element $\boldsymbol{x} \in \mathbb{R}^3$ represents a point coordinate in the 3D space. We first develop an unsupervised **semantic pooling**. After applying ground removal to eliminate redundant ground points, our critical observation shows that most objects become isolated in the 3D space. Consequently, we utilize DBSCAN (Ester et al., 1996) to cluster the remaining points, filtering out clusters that are too large or high to be semantically meaningful using simple yet effective heuristics. Unlike the *bottom-up* proposal generation method that blindly clusters points with a predefined distance (ball query) or number ($k$NN), such as ProposalContrast

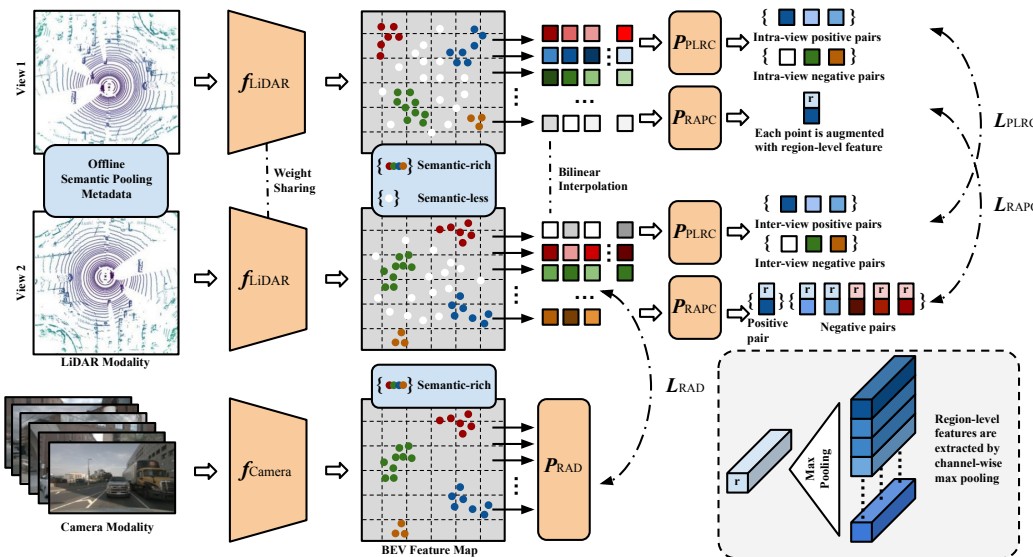

Figure 1: Illustration of our *CALICO* framework, where $P_{\text{PLRC}}$, $P_{\text{RAPC}}$, and $P_{\text{RAD}}$ denote the projectors. $f_{\text{LiDAR}}$ is firstly pretrained by PRC using $L_{\text{PLRC}}$ and $L_{\text{RAPC}}$. $f_{\text{Camera}}$ is then pretrained by contrastive distillation using $L_{\text{RAD}}$.

(§ 2.1) which may lose object-level semantics, our semantic pooling is performed in a *top-down* manner to better capture object-level information. We augment every point with its belonging region in the 4-th dimension, forming $\mathcal{X}' = \{\boldsymbol{x}' \in \mathbb{R}^4\}$. We denote the 4-th dimension, *i.e.*, $\boldsymbol{x}'[3] = -1$ and $\boldsymbol{x}'[3] \in \mathbb{Z}_0^+$ as semantic-less and semantic-rich points, respectively. The value in a semantic-rich point represents which semantic cluster it belongs to.

Subsequently, we apply two sets of spatial augmentations to $\mathcal{X}'$, resulting in $\mathcal{X}'^1 = T_1(\mathcal{X}')$ and $\mathcal{X}'^2 = T_2(\mathcal{X}')$. The augmentation, a combination of randomly sampled rotation, flip, and scaling, is only applied to the LiDAR input coordinates. We then employ our **RPC** (§ 2.3) to train the LiDAR backbone $f_{\text{LiDAR}}$. Given the image input $\boldsymbol{X} \in \mathbb{R}^{N \times H \times W}$, where $N$ represents the number of cameras, we use the camera backbone $f_{\text{Camera}}$ to generate the feature map $\boldsymbol{F}^{\text{Camera}}$. Finally, we apply **RAD** (§ 2.4) to $\boldsymbol{F}^{\text{Camera}} = f_{\text{Camera}}(\boldsymbol{X})$ and $\boldsymbol{F}^{\text{LiDAR}} = f_{\text{LiDAR}}(\mathcal{X})$ to train $f_{\text{Camera}}$.

## 2.3 POINT-REGION CONTRAST

**Point-Level Region Contrast**. PLRC (Bai et al., 2022b) is the SOTA-pertaining method for 2D object detection, which partitions an image into grids and conducts intra- and inter-image contrastive learning. We adapt PLRC to 3D object detection for LiDAR point clouds. Specifically, we sample $N$ corresponding semantic-rich points that belong to $N_{\mathcal{R}}$ regions and $M$ semantic-less points in $\mathcal{X}'^1$ and $\mathcal{X}'^2$. We use $r_i = r_j$ to show that point $i$ and $j$ belong to the same region We further leverage bilinear interpolation $BI()$ to extract the $N + M$ point-level embedding from the feature map. An MLP projector is attached to $f_{\textbf{LiDAR}}$ to generate features for contrastive learning, *i.e.*, $\{\boldsymbol{z}_i^1\}_{i=1}^{N+M} = \{P_{\text{PLRC}}(BI(\boldsymbol{F}^{1^{\text{LiDAR}}}, i))\}_{i=1}^{N+M}$. Different from (Bai et al., 2022b), we leverage the semantic-less points to enrich the negative pairs, which relieves the class collision problems, where the objective is formulated as:

$$L_{\text{PLRC}} = -\frac{1}{N} \sum_i \frac{1}{C} \sum_{r_i = r_j} \log \frac{\exp(\boldsymbol{z}_i^1 \cdot \boldsymbol{z}_j^2 / \tau)}{\sum_{j=1}^{N+M} \exp(\boldsymbol{z}_i^1 \cdot \boldsymbol{z}_j^2 / \tau)}, \tag{1}$$

where $C$ is a normalization factor that denotes the number of positive pairs. The key improvements are from our finegrained semantic pooling and novel design to enrich the negative pairs, which significantly distinguish our proposal. We find that our PLRC could boost the performance of downstream tasks when the finetuning data is limited. However, due to the imperfect region assignment, PLRC is prone to overfitting when the finetuning dataset becomes larger, which is detailed in § 3.

**Point-Region Contrast**. We further improve our design by introducing a new region-aware point contrast (RAPC) scheme. Different from ProposalContrast that uses complex cross-attention modules, we concatenate region-level features with the point embedding. Specifically, we leverage another

MLP projector to generate $\{\boldsymbol{p}_i^1\}$ and $\{\boldsymbol{p}_i^2\}$. The region feature is extracted by channel-wise max pooling $\boldsymbol{p}_r = \mathrm{MaxPool}\{\boldsymbol{p}_i | i \in r\}$ and the final representation is point feature $\boldsymbol{p}_i = [\boldsymbol{p}_i; \boldsymbol{p}_r]$. The objective is then formulated as:

$$L_{\mathrm{RAPC}} = -\frac{1}{N} \sum_i \log \frac{\exp(\boldsymbol{p}_i^1 \cdot \boldsymbol{p}_i^2 / \tau)}{\sum_{j=1}^{N+M} \exp(\boldsymbol{p}_i^1 \cdot \boldsymbol{p}_j^2 / \tau)}, \quad L_{\mathrm{PRC}} = \alpha L_{\mathrm{PLRC}} + (1-\alpha)L_{\mathrm{RAPC}} \quad (2)$$

PLRC, fundamentally a region contrast, emphasizes consistent representation learning within the same region. Conversely, RAPC aims for equivalent global representation learning with regional awareness. Both objectives balance localization and object-level semantics for improved overall performance when combined.

## 2.4 REGION-AWARE DISTILLATION

The second module of *CALICO* is region-aware LiDAR-to-camera distillation (RAD). As introduced in § 2.1, LiDAR-to-camera distillation has been intensively studied in the past year due to the rapid progress in 2D and 3D BEV perception. However, prior works suppose an existing expertly-trained teacher model from the LiDAR modality. In our setup, the LiDAR backbone is trained with our PRC in a self-supervised manner. Therefore, the feature map from the LiDAR backbone cannot be viewed as an *oracle*. Drawing inspiration from BEVDistill (Chen et al., 2022b) and contrastive distillation (Tian et al., 2019), we propose a distillation scheme that operates under fully self-supervised learning conditions. Our approach intuitively focuses on distilling features within our semantically pooled area in a contrastive manner, as these features have been well-learned through PRC. Moreover, we introduce a paradigm to achieve regional awareness. A key observation here is that our unsupervised semantic pooling may generate numerous meaningful foreground objects, but the number of points within a region is nondeterministic. Consequently, smaller but critical objects are less weighted, and there are objects like buildings or bushes that contain many points but lack the information we are interested in. In RAD, we assign region-wise weight to every point-level embedding. The loss function is thus formulated as:

$$L_{\mathrm{RAD}} = -\frac{1}{N_{\mathcal{R}}} \sum_{\mathcal{S} \in \mathcal{R}} \frac{1}{N_{\mathcal{S}}} \sum_{i \in \mathcal{S}} \log \frac{\exp(\boldsymbol{c}_i \cdot \boldsymbol{l}_i / \tau)}{\sum_j \exp(\boldsymbol{c}_i \cdot \boldsymbol{l}_j / \tau)} \quad (3)$$

where $N_{\mathcal{R}}$ and $N_{\mathcal{S}}$ are the number of pooled regions and sampled points in the region $\mathcal{S}$, respectively. $\boldsymbol{l}$ denotes the interpolated feature from $\boldsymbol{F}^{\mathrm{LiDAR}}$ directly and $\boldsymbol{c}$ is projected feature from $\boldsymbol{F}^{\mathrm{Camera}}$. As shown in the formulation, the weight for each point-level feature is normalized based on the number of points $\mathcal{S}$. Different from existing studies that generate center-based masks for groundtruth objects (Chen et al., 2022b), we treat every point in one region the same as the region assignments are from heuristics.

## 3 EXPERIMENTS AND RESULTS

In this section, we introduce the evaluation of *CALICO* with a breakdown of contributions from different components. We first describe the experimental setup in § 3.1 and detail the evaluation results in § 3.2 and § 3.3. Lastly, we conduct a comprehensive analysis and ablation studies of *CALICO* in § 3.4 and § 3.5.

## 3.1 EXPERIMENTAL SETUPS

**Dataset and Tasks**. We adopt the widely used experimental setups for *CALICO*, *i.e.*, first pretraining the backbone with massive unlabeled data and then fine-tuning the model on a significantly smaller amount of annotated data. We utilize the nuScenes dataset (Caesar et al., 2020) to evaluate our method. This large-scale self-driving dataset is released under the CC BY-NC-SA 4.0 license and has been employed in the settings of BEVFusion (Liu et al., 2023). The nuScenes dataset provides diverse annotations to support various tasks, such as 3D object detection/tracking and BEV map segmentation. Each of the 40,157 annotated samples includes six monocular camera images with a $360°$ field of view (FoV) and a 32-beam LiDAR scan. In our study, we aim to tackle both 3D object detection and BEV map segmentation tasks. Our 3D object detection task focuses on 10 foreground classes and the BEV map segmentation task considers 6 background classes. We additionally incorporate the Waymo dataset for 3D object detection, adhering to the common protocol outlined in (Yin et al., 2022). We finetune our *CALICO* pretrained model using 20% of labeled examples from the training

Table 1: NuScenes Evaluation Results of *CALICO* with Baselines on the 3D Object Detection Task. Rand. Init. (C) means that only $f_{\text{Camera}}$ is randomly initialized and $f_{\text{LiDAR}}$ is pretrained by PRC.

| Training Data | Method | Modality | NDS ↑ | mAP ↑ |
|---|---|---|---|---|
| 5% | Rand. Init. | L | 37.4 | 33.1 |
| | PointContrast | L | 43.0 | 36.7 |
| | ProposalContrast | L | 43.1 | 37.0 |
| | PLRC (Ours) | L | **46.3** | **38.2** |
| | **PRC (Ours)** | L | 46.0 | 38.2 |
| | **PRC**+Rand. Init. (C) | L+C | 46.1 | 40.2 |
| | SimIPU | L+C | 45.8 | 39.1 |
| | **PRC**+BEVDistill | L+C | 47.5 | 41.0 |
| | *CALICO* **(Ours)** | L+C | **47.9** | **41.7** |
| 10% | Rand. Init. | L | 48.0 | 41.1 |
| | PointContrast | L | 51.2 | 42.3 |
| | ProposalContrast | L | 51.1 | 42.1 |
| | PLRC (Ours) | L | 51.9 | 43.3 |
| | **PRC (Ours)** | L | **53.1** | **44.1** |
| | **PRC**+Rand. Init. (C) | L+C | 52.9 | 48.9 |
| | SimIPU | L+C | 52.4 | 47.5 |
| | **PRC**+BEVDistill | L+C | 53.6 | 49.7 |
| | *CALICO* **(Ours)** | L+C | **53.9** | **50.0** |
| 20% | Rand. Init. | L | 56.7 | 47.1 |
| | PointContrast | L | 57.5 | 48.3 |
| | ProposalContrast | L | 57.4 | 48.0 |
| | PLRC (Ours) | L | 57.8 | 48.6 |
| | **PRC (Ours)** | L | **58.9** | **49.5** |
| | **PRC**+Rand. Init. (C) | L+C | 59.0 | 54.0 |
| | SimIPU | L+C | 58.9 | 53.4 |
| | **PRC**+BEVDistill | L+C | 59.2 | 54.4 |
| | *CALICO* **(Ours)** | L+C | **59.5** | **54.8** |
| 50% | Rand. Init. | L | 61.0 | 53.2 |
| | PointContrast | L | 61.4 | 53.5 |
| | ProposalContrast | L | 61.0 | 53.1 |
| | PLRC (Ours) | L | 60.8 | 53.2 |
| | **PRC (Ours)** | L | **62.1** | **54.1** |
| | **PRC**+Rand. Init. (C) | L+C | 62.1 | 58.9 |
| | SimIPU | L+C | 62.0 | 58.6 |
| | **PRC**+BEVDistill | L+C | 62.3 | 59.6 |
| | *CALICO* **(Ours)** | L+C | **62.7** | **60.1** |

set using 30 epochs and evaluate the performance on the validation set. We detailed the task-specific settings in § 3.2 and § 3.3.

**Network Architectures and Implementations**. We adopt the architecture of BEVFusion (Liu et al., 2023) throughout our evaluation. In our main evaluation, we use the PointPillars (Lang et al., 2019) backbone for $f_{\text{LiDAR}}$ and Swin-T (Liu et al., 2021) for $f_{\text{Camera}}$. An LSS (Philion & Fidler, 2020) view transformer is prepended to $f_{\text{LiDAR}}$ to transform the image from the perspective to BEV space. We by default leverage CenterPoint (Yin et al., 2021) and ConvNet heads for 3D object detection and BEV map segmentation. Following (Liu et al., 2023; Lang et al., 2019), we aggregate each LiDAR point cloud with up to 10 consecutive frames, which is a standard procedure for nuScenes. The DBSCAN in our semantic pooling has a minimum of 5 points and a distance of 0.75 meters for clustering. We implement all three projectors with two linear layers, where only the first layer consists of batch normalization and ReLU layers. The output dimension of the projectors is set as 128. Our view augmentation $T()$ includes random rotation of $[-90°, 90°]$, random scaling of $[0.9, 1.1]$, and random flipping along the X or Y axis. The temperature factor in Equations 1 and 2 are set $\tau = 0.07$. During pretaining, we sample $N = 1024$ semantic-rich and $M = 1024$ semantic-less points and set $\alpha = 0.5$. We pretrain the $f_{\text{LiDAR}}$ and $f_{\text{Camera}}$ using PRC and RAD both for 20 epochs on the entire training set. We leverage $\{5\%, 10\%, 20\%, 50\%\}$ of the training set with annotations to further finetune the model with detection or segmentation head attached for another 20 epochs. Other baselines are trained with the same number of epochs for fair comparisons. All experiments are conducted on 4 V100 GPUs with 32GB memory (v10, 2023).

### 3.2  3D Object Detection Evaluation

**Settings**. Our main metrics for nuScenes evaluation are mean average precision (mAP) and nuScenes distance error (NDS). For Waymo evaluation, we follow the existing studies to report the Level-

2 average precision (AP) and APH, a customized metric defined by Waymo (Sun et al., 2020b), incorporating the heading information.

**NuScenes Results**. Table 1 presents a comparison of the evaluation results for *CALICO* and other baseline methods in the context of 3D object detection. Notably, PRC achieves state-of-the-art (SOTA) results for LiDAR-only perception. Compared to the randomized initialization baseline, PRC demonstrates an improvement of **8.6%** and **5.1%** in mAP and NDS, respectively, when fine-tuning with 5% of the training data. Furthermore, PRC surpasses PointContrast and ProposalContrast by a considerable margin. The performance gains of PRC do not diminish as the fine-tuning data increases, such as when 50% of the labeled data is utilized; in this case, PRC achieves enhancements of 1.1% and 0.9% in mAP and NDS compared to the baseline method. It is worth noting that region-focused approaches, including ProposalContrast and our PLRC, tend to be susceptible to overfitting due to the imperfect assignment of regions from heuristics. Nevertheless, PRC generally outperforms other methods across 5 additional metrics, albeit with minor fluctuations. Additionally, *CALICO*, *i.e.,* PRC+RAD, achieves SOTA performance in 3D object detection under multimodal settings, where outperforms PRC by 1.8% and 1.5% in mAP and NDS. Although SimIPU outperforms the vanilla randomized initialization, it cannot beat PRC with randomly initialized $f_{\text{Camera}}$. As introduced before, the transformation from the perspective space to BEV is implicit, hindering the optimization for $f_{\text{LiDAR}}$. In contrast, our two-stage optimization enables more stable pretraining for both $f_{\text{Camera}}$ and $f_{\text{LiDAR}}$. RAD consistently exhibits tangible improvements over BEVDistill, primarily due to our specific optimization for the semantically pooled regions. Note that all the BEVDistill results are based on our PRC for pretrained $f_{\text{LiDAR}}$, and the camera-only results can be found in Appendix B.

Table 2: Waymo Evaluation Results of *CALICO* with Baselines on the 3D Object Detection Task.

| Training Data | Method | Modality | AP ↑ | APH ↑ |
|---|---|---|---|---|
| 20% | Rand. Init. | L | 63.2 | 61.0 |
| | PointContrast | L | 65.2 | 62.6 |
| | ProposalContrast | L | 66.3 | 63.7 |
| | **PRC (Ours)** | L | **68.6** | **65.5** |
| | SimIPU | L+C | 68.4 | 65.5 |
| | *CALICO* **(Ours)** | L+C | **71.6** | **68.0** |

**Waymo Results**. We present our experimental results on the Waymo 3D object detection benchmark in Table 2. As shown, our PRC consistently outperforms the previous SOTA method, ProposalContrast, registering a more pronounced improvement on the Waymo dataset compared to nuScenes when only pretrained using the LiDAR modality. Additionally, *CALICO* also achieves the best results in performance for sensor-fusion perception, which are 8.4% and 3.2% improvements than the baseline and previous SOTA methods, respectively. The results fairly demonstrate the generalization of our proposal on different benchmarks.

Table 3: Cross-Dataset Evaluation Results of *CALICO* with Baselines on the 3D Object Detection Task.

| Training Data | Method | Modality | NDS ↑ | mAP ↑ |
|---|---|---|---|---|
| 10% | Rand. Init. | L | 48.0 | 41.1 |
| | PointContrast | L | 47.9 | 41.1 |
| | ProposalContrast | L | 48.5 | 41.8 |
| | **PRC (Ours)** | L | **50.6** | **42.7** |
| | SimIPU | L+C | 50.8 | 44.7 |
| | *CALICO* **(Ours)** | L+C | **51.9** | **47.9** |

**Cross-Dataset Results**. We also report the evaluation results of our cross-data experiment. Specifically, we leverage the backbones pretrained on the Waymo dataset and finetune them on the 10% data setting in nuScenes. As Table 3 presents, our PRC and *CALICO* consistently deliver the best performance, outperforming the previous methods by a significant margin, which further illustrates the effectiveness of our proposal.

## 3.3 BEV MAP SEGMENTATION EVALUATION

**Settings**. We use the intersection-over-union (IoU) and mean IoU as the main metrics in this evaluation. we evaluate the binary segmentation performance for every class and select the highest IoU across different thresholds (Xu et al., 2022). The target area is a $[-50, 50] \times [-50, 50]$ m$^2$ plane in the ego vehicle's coordinate, following prior studies (Philion & Fidler, 2020; Xu et al., 2022). We adopt other settings in BEVFusion to jointly perform binary segmentation for all classes to accelerate the training and inference.

**Results**. Table 4 presents the evaluation results of BEV map segmentation. We only evaluate the pretraining methods for multimodal perception as the presence of texture information in the multiview images is essential for this task (Liu et al., 2023). Our *CALICO*, consistently outperforms other methods in terms of mIoU across most categories. Notably, *CALICO* shows an improvement of

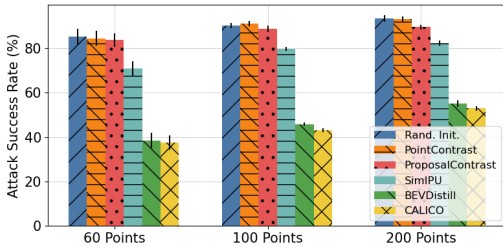 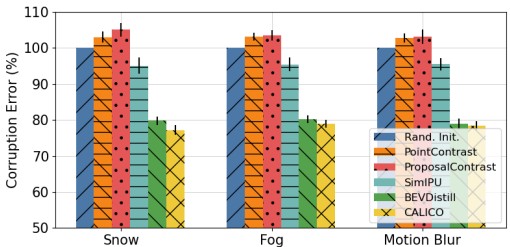

Figure 2: Attack Success Rates (↓) of Black-box Adversarial Attacks with Various Spoofing Points on Models Pretrained with Different Methods.

Figure 3: Corruption Errors (↓) of Different Corruption Types on Models Pretrained with Different Methods.

5.7% over baseline approaches when finetuning on the 5% of the training set. The improvement, 1.3%, remains tangible when finetuning 50% of the training data. Although SimIPU beats the model finetuned from scratch, it still cannot perform better than PRC trained $f_{\text{LiDAR}}$ with randomly initialized $f_{\text{Camera}}$. As the target area typically belongs to the background—less the focus compared to objects like vehicles and pedestrians—the performance enhancement from pretraining is not as pronounced as in the 3D object detection task, though we are the first to study the pretraining framework for BEV map segmentation.

Table 4: IoU (↑) Evaluation Results of *CALICO* with Baseline Methods on BEV Map Segmentation.

| Training Data | Method | Drivable | Ped. Cross | Walkway | Stop Line | Carpark | Driver | Average |
|---|---|---|---|---|---|---|---|---|
| 5% | Rand. Init. (L+C) | 67.5 | 33.2 | 43.9 | 19.1 | 24.3 | 30.0 | 36.3 |
| | **PRC**+Rand. Init. (C) | 70.1 | 36.0 | 45.8 | 23.5 | 26.9 | 31.6 | 39.0 |
| | SimIPU | 68.9 | 35.5 | 45.8 | 22.9 | 26.6 | 31.5 | 38.5 |
| | **PRC**+BEVDistill | 72.1 | 37.9 | 48.0 | 24.5 | 29.3 | 33.3 | 40.9 |
| | *CALICO* (Ours) | **73.1** | **39.2** | **49.3** | **26.0** | **30.6** | **33.8** | **42.0** |
| 10% | Rand. Init. (L+C) | 73.2 | 40.6 | 50.4 | 28.5 | 34.9 | 35.2 | 43.8 |
| | **PRC**+Rand. Init. (C) | 74.8 | 42.0 | 51.4 | 30.6 | 36.3 | 36.8 | 45.3 |
| | SimIPU | 74.4 | 41.8 | 51.2 | 30.6 | 36.0 | 36.5 | 45.1 |
| | **PRC**+BEVDistill | 75.8 | 43.1 | 52.0 | 31.8 | 37.5 | 38.1 | 46.4 |
| | *CALICO* (Ours) | **76.5** | **44.0** | **53.1** | **32.5** | **38.8** | **39.0** | **47.3** |
| 20% | Rand. Init. (L+C) | 77.3 | 49.2 | 56.1 | 34.4 | 43.1 | 39.8 | 50.0 |
| | **PRC**+Rand. Init. (C) | 78.2 | 50.4 | 57.0 | 35.6 | 44.8 | 40.5 | 51.1 |
| | SimIPU | 78.0 | 50.4 | 56.7 | 35.4 | 44.6 | 40.2 | 50.9 |
| | **PRC**+BEVDistill | 78.8 | 50.9 | 57.7 | 36.0 | 45.3 | 41.1 | 51.6 |
| | *CALICO* (Ours) | **79.4** | **51.5** | **58.3** | **36.7** | **45.9** | **41.9** | **52.3** |
| 50% | Rand. Init. (L+C) | 81.1 | 55.2 | 61.8 | 40.2 | 50.4 | 43.7 | 55.4 |
| | **PRC**+Rand. Init. (C) | 81.6 | 55.8 | 62.3 | 41.0 | 50.9 | 44.4 | 56.0 |
| | SimIPU | 81.6 | 55.5 | 62.2 | 40.8 | 51.0 | 44.3 | 55.9 |
| | **PRC**+BEVDistill | 82.0 | 56.5 | 62.7 | 41.3 | **51.3** | 44.5 | 56.4 |
| | *CALICO* (Ours) | **82.4** | **57.0** | **63.1** | 41.3 | 51.6 | **44.9** | **56.7** |

## 3.4 ROBUSTNESS EVALUATION

In this section, we conduct two sets of robustness evaluations, *i.e.,* LiDAR spoofing attacks and common corruptions, of *CALICO* with other baseline methods.

**Adversarial Robustness**. 3D object detection is known to be vulnerable to LiDAR spoofing attacks (Sun et al., 2020a), where adversaries are capable of physically injecting malicious points into the LiDAR sensors to create driving hazards like emergency braking and steering. In this section, we leverage the black-box adversarial attacks in (Sun et al., 2020a) to evaluate the robustness of the models pretrained by *CALICO* and other methods. Specifically, we leverage attack traces with 60, 100, and 200 points to spoof fake objects 10 meters in front of the ego vehicle. We follow the same settings in (Sun et al., 2020a) to set up the digital attacks. As the black-box attack requires models that are ready to be deployed on the road, we choose models fine-tuned on the 50% training data which deliver satisfactory performance. Figure 2 shows the attack success rates (ASR) on different methods. We find that *CALICO* is able to reduce the ASR by 45.3% on average compared to models trained from scratch. A similar performance is achieved by BEVDistill as it depends on our PRC trained $f_{\text{LiDAR}}$. Our results consolidate the findings in (Sun et al., 2020a), where the 3D LiDAR modality dominates the 3D object detection model during standard training. *CALICO* helps to achieve more balanced pretraining on both modalities.

**Corruption Robustness**. Autonomous vehicles inherently face challenges posed by shifts in the distribution of input data, such as changes in weather conditions. In light of recent findings from nuScenes-C (Kong et al., 2023), we evaluate the robustness of various pretraining methods in the face of such corruptions by distorting the LiDAR input with snow, fog, and motion blur. This experiment employs models fine-tuned on 10% of the training data. The mean corruption error (mCE) of each model is depicted in Figure 3. For the calculation of mCE, we establish the model trained from scratch as the baseline (100%). Detailed information regarding the experimental setup is provided in Appendix B. The results reveal that *CALICO* outperforms all other baseline models, achieving the lowest mCE at 78.2%. It is worth noting that both PointContrast and ProposalContrast only pretrain $f_{\text{LiDAR}}$, thereby overemphasizing this modality, which results in higher mCEs than those trained from scratch.

## 3.5 ABLATION STUDIES

Table 5: Ablation Study of $\alpha$ in PRC.

| $\alpha$ | 5% NDS | 5% mAP | 50% NDS | 50% mAP |
|---|---|---|---|---|
| 0.1 | 44.8 | 37.2 | **62.2** | **54.4** |
| 0.25 | 45.2 | 37.6 | 62.2 | 54.3 |
| 0.5 | 46.0 | 38.2 | 62.1 | 54.1 |
| 0.75 | 46.1 | 38.1 | 61.4 | 53.6 |
| 0.9 | **46.3** | **38.2** | 60.8 | 53.2 |

**Architecture**. In our exploration, we employ sparse VoxelNet (Yan et al., 2018) as $f_{\text{LiDAR}}$ and TransFusion (Bai et al., 2022a) as the detection head. This approach allows us to assess the adaptability of our PRC across diverse architectures and heads. For the evaluation of the downstream 3D object detection task, we use 5% and 10% of the training set. As depicted in Figure 4, our PRC approach demonstrates a universal improvement in mAP and NDS. Notably, the sparse VoxelNet with the TransFusion head consistently outperforms the PointPillars with the CenterPoint head, exhibiting an average improvement of 2.3%. This superior performance, however, comes with a trade-off in inference speed, attributable to the increased complexity of the model.

**Hyperparameters**. We investigate the impact of $\alpha$ in Eq.2. Our findings suggest that $\alpha$ serves as a balance between the performance of PRC on limited and ample labeled finetuning data. As delineated in Table 5, the enhancement on 5% finetuning data is markedly pronounced when $\alpha = 0.9$. Conversely, an $\alpha = 0.1$ effectively mitigates the overfitting issue on 50% of the finetuning data. PLRC prioritizes region-level representation learning, which could supply more valuable heuristics when the finetuning data are scarce. However, during pretraining, the model may excessively fit into the assigned region. As the volume of annotated data increases,

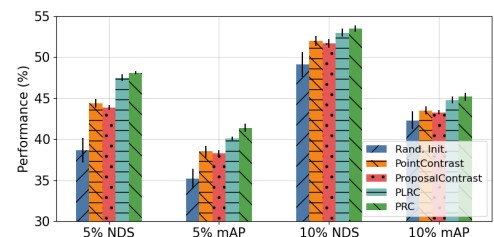

Figure 4: NDS and mAP ($\uparrow$) of 3D Object Detection on the model architecture: $f_{\text{LiDAR}}$ =VoxelNet with the Transfusion head.

PLRC independently falls short compared to the baseline method. In contrast, RAPC emphasizes global invariant representation learning, which can help alleviate the overfitting problem in PRC. In the meanwhile, $\alpha = 0.5$ achieves a better balance between the detection performance on both limited and sufficient annotated data.

## 4 CONCLUSION

In conclusion, we have introduced *CALICO*, a novel pretraining framework for multimodal BEV perception in this paper. *CALICO* applies contrastive objectives on both LiDAR and camera modalities, including two innovative stages: point-region contrast (PRC) and region-aware distillation (RAD). It significantly outperforms baseline methods, with improvements of 10.5% and 8.6% on NDS and mAP, respectively. Additionally, it boosts the robustness of multimodal 3D object detection against adversarial attacks and common corruptions. The flexibility of *CALICO* allows for adaptation to various backbones and heads as well, rendering it a promising approach for multimodal BEV representation learning.

## ACKNOWLEDGMENT

We thank our area chairs and anonymous reviewers for their insightful comments and feedback. This work was supported by NSF under the National AI Institute for Edge Computing Leveraging Next Generation Wireless Networks, Grant # 2112562, NSF Grants CMMI-2038215 and CNS-1930041, and U.S. Department of Homeland Security under Grant Award # 17STQAC00001-06-00.

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

# A    RELATED WORK

In this section, we review two topics related to our study: autonomous driving perception and contrastive pretraining for deep learning.

**Autonomous Driving Perception**. Perception remains a cornerstone of autonomous vehicles, primarily facilitated through LiDAR and camera sensors. LiDAR-based 3D object detection can be classified into point-wise (Shi et al., 2019; 2021; 2020; 2023) and voxel-based (Lang et al., 2019; Yan et al., 2018; Zhou et al., 2020; Zhou & Tuzel, 2018; Choy et al., 2019) representations. The two-stage PointRCNN (Shi et al., 2019) initially segments the object from the point cloud before estimating the bounding box. Despite directly extracting lossless features, point-wise methods are computationally expensive (Geiger et al., 2013). To counter this, voxel-based methods like VoxelNet (Zhou & Tuzel, 2018) and PointPillars (Lang et al., 2019) transform point clouds into BEV pseudo-images, suitable for 2D ConvNets (Liu et al., 2022) and ViT (Dosovitskiy et al., 2020) processing. Detection heads like CenterPoint (Yin et al., 2021) have also been introduced for enhanced end-to-end performance. Camera-based BEV perception methods can be divided into two categories based on their depth estimation approach. Models like BEVDet (Huang et al., 2021) and BEVDepth (Li et al., 2022c) explicitly estimate depth, transforming perspective views into BEV using a dedicated depth estimation branch (Philion & Fidler, 2020). BEVerse (Zhang et al., 2022b) expands on this with a unified multi-task learning framework. In contrast, DETR3D (Wang et al., 2022) and ORA3D (Roh et al., 2022) utilize the DETR (Carion et al., 2020) framework to represent 3D objects as queries and perform cross-attention via a Transformer decoder. BEVFormer (Li et al., 2022e) and PolarFormer (Jiang et al., 2022) introduce novel methods for feature extraction and 3D target prediction. As advancements in both modalities of BEV perception continue, sensor fusion methods such as BEVFusion (Liu et al., 2023) have been proposed to further enhance performance. UVTR (Li et al., 2022b) generates a unified representation in the 3D voxel space. Meanwhile, query-based methods like FUTR3D (Chen et al., 2022a) use 3D reference points as queries and sample features directly from the coordinates of projected planes. Transfusion (Bai et al., 2022a) employs a two-stage pipeline, initially generating proposals using LiDAR features, then refining them by querying image features.

**Contrastive Pretraining for Deep Learning**. Contrastive pretraining has emerged as a significant advancement in the field of deep learning, particularly in self-supervised learning scenarios. It has shown remarkable success across various tasks, demonstrating its effectiveness in learning meaningful representations without relying on labeled data. One of the earliest applications of contrastive learning is the *word2vec* model by Mikolov *et al.* (Mikolov et al., 2013), which uses a contrastive loss function to learn word embeddings, which has inspired a new wave of research in contrastive learning for other data modalities, including images (Park et al., 2020; He et al., 2020; Chen et al., 2020b; Grill et al., 2020; Bai et al., 2022b; Wu et al., 2023), audio (Wang & Oord, 2021; Saeed et al., 2021; Manocha et al., 2021), and 3D data (Xie et al., 2020; Yin et al., 2022; Liang et al., 2021). In the realm of computer vision, contrastive learning has been studied extensively. He *et al.* (He et al., 2020) introduced the Momentum Contrast (MoCo) for unsupervised visual representation learning, which constructs a dynamic dictionary with a queue and a moving-averaged encoder. Chen *et al.* (Chen et al., 2020a) proposed the SimCLR framework, which uses simple data augmentations to learn visual representations effectively. Contrastive pretraining has also been applied to multimodal learning, where the goal is to learn a joint representation of different data modalities. For instance, CLIP by Radford et al. (Radford et al., 2021) applies contrastive learning to connect images and text in a shared embedding space. GLIP (Li et al., 2022a; Zhang et al., 2022a) further extended CLIP to 2D object detection. In the context of 3D point clouds, contrastive learning has been used to learn representations from LiDAR data. Works including PointContrast (Xie et al., 2020) and ProposalContrast (Yin et al., 2022) have utilized contrastive pretraining for 3D object detection tasks, which are detailed in § 2.1.

# B    EXPERIMENTS

**Detailed Experimental Setups**. In this section, we delineate the comprehensive experimental setups employed in our research. For the Sparse VoxelNet and PointPillars backbones, we have utilized voxel sizes of $[0.075, 0.075, 0.2]$ and $[0.2, 0.2, 8]$ respectively, and each voxel is regulated to contain a maximum of 10 points. The ensuing feature maps for $\boldsymbol{F}^{\text{LiDAR}}$ and $\boldsymbol{F}^{\text{Camera}}$ are sized $[180, 180, 256]$ and $[180, 180, 80]$, respectively. To derive maximal advantage from multimodal pretraining, we have

Table 7: Ablation Studies on RAD and *CALICO* with Camera-only Methods.

|  | Method | Modality | NDS ↑ | mAP ↑ |
|---|---|---|---|---|
| 10% | Original PLRC | L | 51.1 | 42.0 |
|  | **PRC (Ours)** | L | **53.1** | **44.1** |
|  | BEVDet | C | 29.5 | 24.1 |
|  | BEVFormer | C | 31.0 | 26.4 |
|  | Vanilla Distillation | C | 31.5 | 26.9 |
|  | Original BEVDistill | C | 33.2 | 27.3 |
|  | **RAD (Ours)** | C | **34.7** | **29.1** |
|  | *CALICO* **(Ours)** | L+C | **53.9** | **50.0** |
| 50% | Original PLRC | L | 60.3 | 52.7 |
|  | **PRC (Ours)** | L | **62.1** | **54.1** |
|  | BEVDet | C | 36.1 | 30.2 |
|  | BEVFormer | C | 36.5 | 30.7 |
|  | Vanilla Distillation | C | 37.0 | 30.9 |
|  | Original BEVDistill | C | 39.0 | 32.3 |
|  | **RAD (Ours)** | C | **40.2** | **34.0** |
|  | *CALICO* **(Ours)** | L+C | **62.7** | **60.1** |

restructured the BEVFusion architecture. Specifically, we employed separate decoders for $f_{\text{LiDAR}}$ and $f_{\text{Camera}}$, and relocated the fusion layer further along the backbone. Additionally, the fusion decoder has been reengineered as a simple convolution layer that merges the feature maps from both modalities. This change ensures the utmost utilization of the pretraining to train a maximum number of parameters and positions the pretrained feature map closer to the heads for various downstream tasks. All projectors feature a middle layer with 256 channels and output a channel dimension of 128. We have observed that the baseline can be enhanced using a higher weight decay of 0.2 to avert overfitting when fine-tuning with a limited amount of annotated data. We employed the AdamW optimizer with a cyclic scheduler and a starting learning rate of $2 \times 10^{-4}$. A gradient maximum clip of 35 was used. All other settings remain consistent with the BEVFusion experiments.

Table 6: Comparison with GCC-3D, RoIAlign-based Constrast, and BEV-MAE. * denotes the results reported in Lin & Wang (2022).

| Method | 5% NDS | 5% mAP | 100% NDS | 100% mAP |
|---|---|---|---|---|
| Rand. Init. | 37.4 | 33.1 | 64.5* | 56.2* |
| GCC-3D | - | - | 65.0* | 57.3* |
| RoIAlign-Contrast | 45.4 | 37.4 | 64.1 | 55.9 |
| BEV-MAE | - | - | **65.1*** | 57.2* |
| PRC | **46.0** | **38.2** | **65.1** | **57.5** |

**Additional Experiments**. We have compared our PRC with a masked-autoencoder pretraining method, BEV-MAE Lin & Wang (2022). Table 6 presents the results of our PRC with other baselines reported in Lin & Wang (2022). It is worth noting that BEV-MAE is not yet published. GCC-3D Liang et al. (2021), on the other hand, requires temporal information to pool semantic areas. We have also compared our PRC with RoIAlign Liang et al. (2021)-based method mentioned in § 2.1. When fine-tuning using 100% of the training data, our PRC model achieves a similar performance increase as GCC-3D and BEV-MAE. However, when fine-tuning with only 5% of the training data, the performance gain with PRC is substantially more pronounced, underlining its superior efficiency in learning from limited labeled data, which is also an important challenge in deep learning Zheng et al. (2023a;b).

Table 8: Ablation Studies on ResNet as $f_{\text{Camera}}$.

|  | Method | NDS ↑ | mAP ↑ |
|---|---|---|---|
| 10% | **PRC**+Rand. Init. (C) | 52.0 | 47.9 |
|  | SimIPU | 51.4 | 46.6 |
|  | **PRC**+BEVDistill | 52.5 | 48.5 |
|  | *CALICO* **(Ours)** | **53.1** | **49.0** |

Table 9: Ablation Studies on Semantic Pooling.

|  | Method | NDS ↑ | mAP ↑ |
|---|---|---|---|
| 10% | Original PLRC | 50.9 | 41.9 |
|  | +Semantic Pooling | 51.5 | 42.8 |
|  | +Negative Sample Augmentation | **51.9** | **43.3** |
| 50% | Original PLRC | 60.0 | 52.5 |
|  | +Semantic Pooling | 60.5 | 53.0 |
|  | +Negative Sample Augmentation | **60.8** | **53.2** |

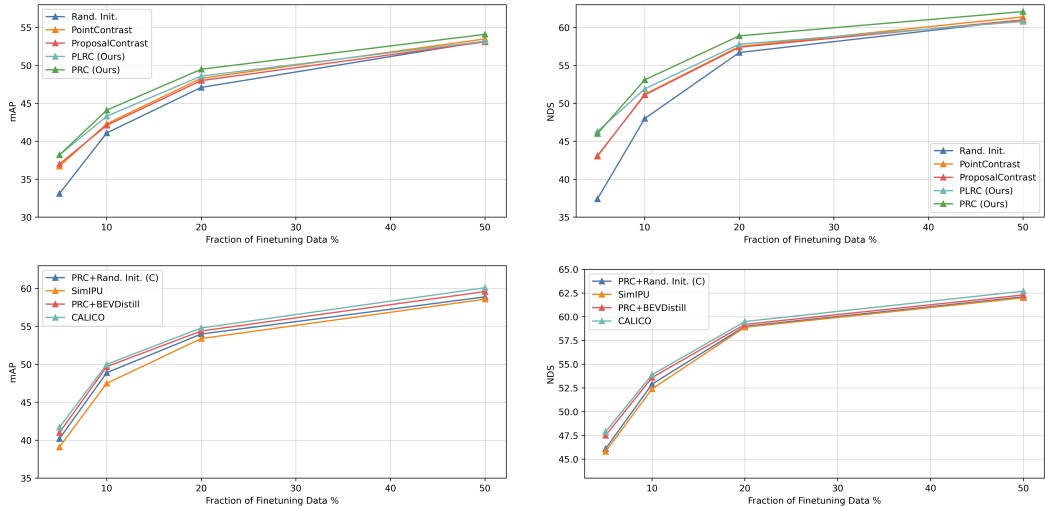

Figure 5: Qualitative Analysis of Evaluation Results in Table 1.

**Comparison with Camera-only Methods**. As shown in Table 7, the improvements of RAD and *CALICO* compared to the original BEVDistill (camera-only) are significant under both 10% and 50% data settings in the nuScenes dataset.

**Ablation on the Camera Backbones**. we have integrated ResNet as an alternative backbone for the camera modality. Specifically, we use our PRC pretrained LiDAR backbone as the teacher model and leverage the 10% finetuning data setting in this experiment. This further step is taken to demonstrate the adaptability and generalizability of our framework across different backbone architectures. As presented in Table 8, our *CALICO* delivers a consistent trend in performance between Swint-T (Liu et al., 2021) and ResNet (He et al., 2016) as image backbones, which consistently achieves the best performance compared to other baseline methods.

**Ablation on Our Semantic Pooling**. We also conducted experiments specifically focusing on the impact and necessity of our negative sample augmentation strategy within our semantic pooling operation. As presented in Table 9, these results clearly demonstrate the effectiveness of our negative sample augmentation method in improving the overall model performance.

## B.1 QUALITATIVE ANALYSIS

As we have presented a significant amount of quantitative results in the main body of our paper, we therefore show some qualitative analysis and insights in this section. In particular, we plot the results of Table 1 with different pretraining methods. As shown in Figure 5, we observe that the benefits of pretraining diminish with the increase in fine-tuning data. This highlights the nuanced trade-off between pretraining and fine-tuning in model performance. Furthermore, when comparing our proposed PRC method (augmented with the RAPC component) to the PLRC approach, it becomes evident that our method demonstrates more significant improvements, especially when finetuning on 50% of the dataset. This finding underscores the efficacy of our approach in scenarios with limited fine-tuning data, an aspect that we believe contributes significantly to the field. Moreover, our ablation study in Table 5 also showcases that the RAPC component in our design serves as a good regularization term to balance the performance of our framework among different levels of availability in finetuning data.

## C VISUALIZATIONS

We visualized both sampled semantic pooling and intermediate feature maps from our *CALICO* framework. As shown in Figures 6 and 9. The feature maps from *CALICO* are sharper than the ones from previous studies.

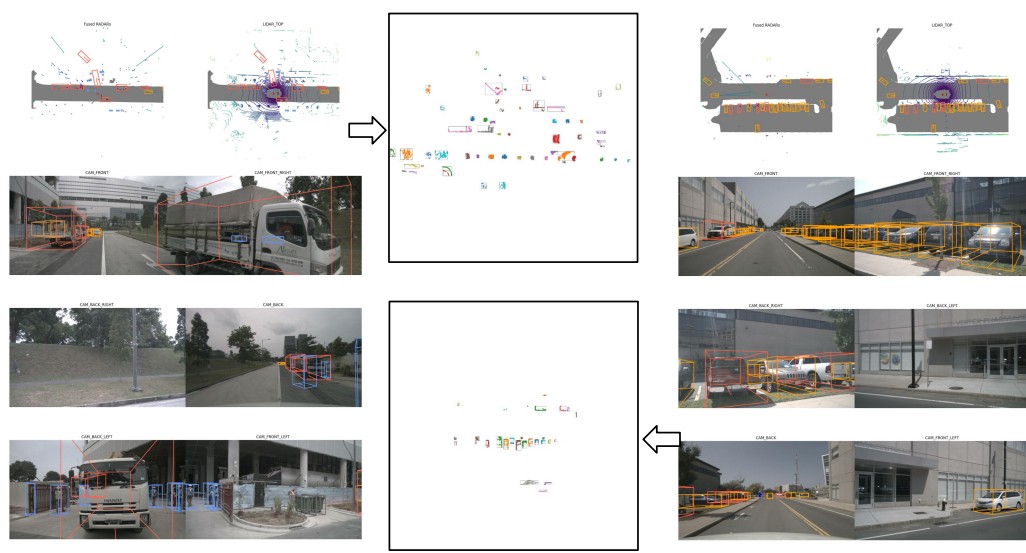

Figure 6: Sampled Visualizations of Our Semantic Pooling. Sub-plots on the two sides represent the ground truth, respectively.

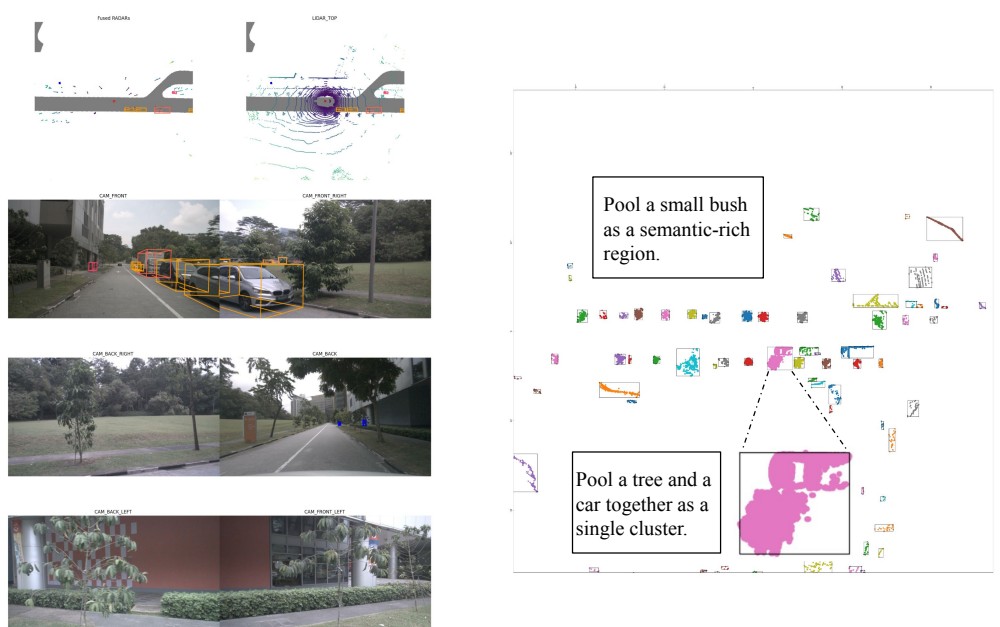

Figure 7: Sampled Visualizations of Failure Cases from Our Semantic Pooling.

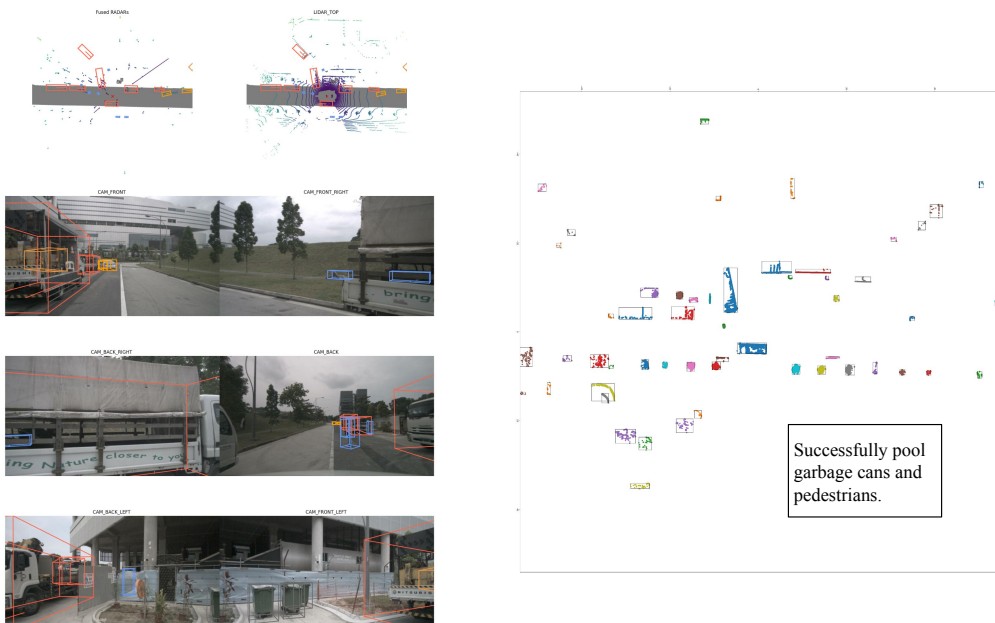

Figure 8: Sampled Visualizations of Special Cases from Our Semantic Pooling.

As shown in Figure 6, we found that our **semantic pooling** could extract most of the objects of interest accurately. While we acknowledge potential errors due to the heuristic nature of unsupervised pooling, we firmly believe that its efficacy meets the requirements of self-supervised pretraining. We also reported the results from vanilla dense distillation methods in Table 7 using the 10% and 50% data settings, due to time constraints. The results fairly demonstrate that RAD significantly improves performance.

In our exploration of the unsupervised nature of our semantic pooling operation in the *CALICO* framework, we encountered certain limitations. For instance, the semantic pooling sometimes erroneously clusters tree points as semantic-rich, as shown in Figure 7. Nonetheless, these instances did not significantly impact overall performance. The primary objective during the pretraining phase is to learn robust and prominent representations within the model's backbone, a process that remains largely independent from downstream tasks. Mispooling a small number of random objects as semantic-rich does not detrimentally affect the contrastive pretraining for the model backbone because the objective is to learn the correspondence between regions. Furthermore, the core concept of semantic pooling is pivotal, transcending the specifics of its implementation. Our current implementation utilized a basic approach to achieve region partitioning in LiDAR BEV space relying on independent LiDAR frames. Enhancements to semantic pooling could include integrating temporal data to refine point aggregation or to better aggregate points and identify moving points. However, such an approach necessitates temporally continuous data, introducing an additional layer of assumption and complexity. *CALICO* is designed to be versatile with respect to the choice of clustering algorithms, as long as the region partitioning is reasonable, as the core of *CALICO* is the following contrastive learning design based on the partitioned regions.

We also visualized the feature map from the camera modality with/without using *CALICO* in Figure 10. As we mentioned in the rebuttal, camera features trained from scratch are not salient to contribute to robustness improvement.

## D  DISCUSSION AND BROADER IMPACT

While our *CALICO* exhibits considerable potential in enhancing the performance of downstream tasks for BEV perception, several challenges persist in this field. One major hurdle is the definition of "positive" and "negative" pairs in the 3D space. We have devised point-wise pairs to balance both region- and scene-level contrasts. Additionally, the implicit transformation from pixel to BEV presents

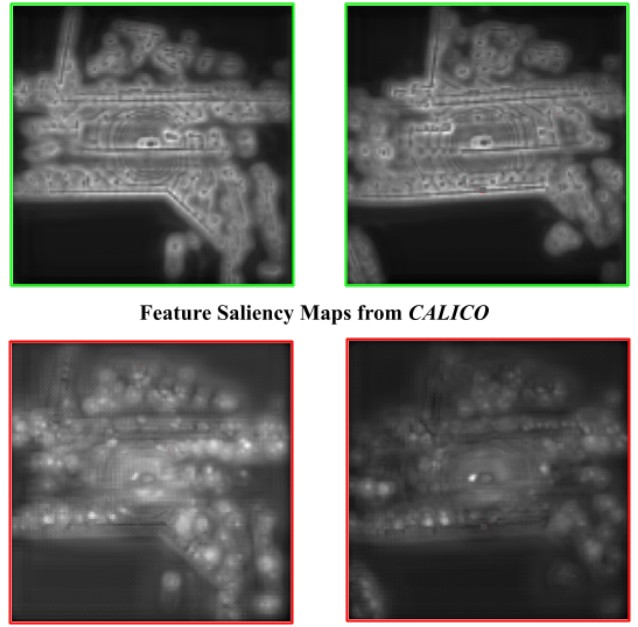

Figure 9: Sampled Visualizations of Intermediate Feature Maps.

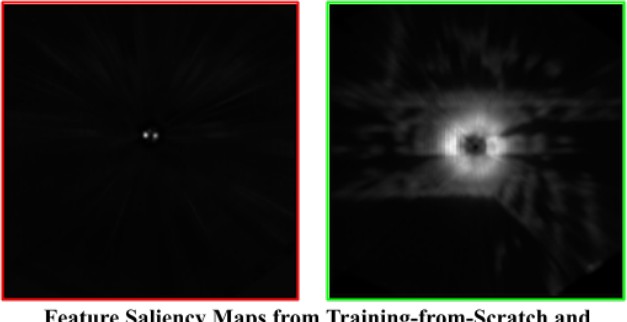

Figure 10: Sampled Visualizations of Camera Feature Maps.

another challenge. We envision a more principled and unified design as a promising area of future research. Furthermore, *CALICO* introduces additional computational and memory consumption, so enhancing efficiency also constitutes a promising future direction.

### D.1 LIMITATION

**Theoretical Understanding**: Our framework, *CALICO*, is based on empirical design and data-driven analysis. The current pretraining methods, however, face significant limitations regarding theoretical understanding. One of the primary issues is that these methods often operate as "black boxes". This opacity makes it challenging to develop a deep theoretical understanding of the underlying mechanics of these algorithms. Another limitation is the reliance on empirical results over theoretical foundations. While pretraining methods have shown remarkable success in various applications, this success is often benchmarked through performance metrics on specific tasks rather than grounded in theoretical principles. This approach can lead to a lack of generalizability, where models perform well on certain types of data or tasks but fail in others. We have diligently endeavored to showcase the effectiveness and generalizability of *CALICO* across a diverse array of benchmarks. Moreover,

the complexity and scale of pretraining models pose a significant hurdle. These models often have millions, if not billions, of parameters, making it incredibly difficult to dissect and understand the role and interaction of each component. This complexity hinders the development of a robust theoretical framework that can predict model behavior under different conditions. We believe that building theoretical frameworks is a promising avenue in this field and empirical contributions like ours can lay the groundwork for future theoretical advancements.

**Robustness against Calibration Errors**: Our current *CALICO* operates under the premise of accurate alignment between LiDAR and camera data. This alignment is pivotal for the efficacy of our pretraining methodology, particularly due to the necessity of data synchronization for successful knowledge distillation in RAD. Therefore, our method demonstrates similar performance to other baseline approaches when we intentionally altered the calibration matrix between the two modalities. On one hand, we acknowledge that in real-world scenarios, misalignments due to calibration errors can occur, potentially impacting the performance of systems like ours. Although our current study did not specifically focus on this aspect, we recognize its importance in the broader context of autonomous driving. On the other hand, evaluating calibration errors is challenging, as misalignments result in differing localization ground truths for both modalities. Therefore, determining a method and metric to quantify the evaluation is controversial.

## D.2 BROADER IMPACT

Our proposed *CALICO* methodology holds substantial potential to improve the safety and efficiency of autonomous driving systems. This broader impact can be articulated along several lines of thought.

**Enhanced Perception Accuracy**: By facilitating a deeper understanding of diverse and complex driving scenarios, the contrastive pretraining method could significantly enhance the perception accuracy of autonomous driving systems. This will allow vehicles to more precisely identify and interpret environmental elements such as pedestrians, other vehicles, traffic signals, and road conditions. Improved accuracy will in turn reduce the likelihood of perceptual errors leading to accidents.

**Reliable Interpretation of Ambiguous Situations**: Contrastive pretraining enables the model to learn more effectively from less labeled data. This will help in interpreting ambiguous situations by generalizing the acquired knowledge. As a result, autonomous vehicles can respond appropriately to unexpected or rare traffic scenarios, further enhancing safety.

**Increased Robustness**: Our proposed method could lead to autonomous driving systems that are more robust to adversarial attacks and natural distribution shifts. By learning a rich, discriminative feature space, the system can better differentiate between genuine signals and potential threats or anomalies. This robustness contributes to an overall increase in system resilience and safety.

