# OpenReview forum: "CALICO: Self-Supervised Camera-LiDAR Contrastive Pre-training for BEV Perception"
_ICLR.cc/2024/Conference — ICLR 2024 poster_

### Official Review · Reviewer_ksX5 · 2023-10-28

**Soundness:** 4 excellent
**Presentation:** 4 excellent
**Contribution:** 4 excellent
**Rating:** 6
**Confidence:** 5

**Summary:**

In this paper, the authors try to fill the hole in pre-training for Camera-LiDAR BEV perception. The authors apply contrastive learning on both LiDAR and camera modalities in two stages. The authors develop point-wise positive and negative pairs to balance both region- and scene-level contrasts.

**Strengths:**

In sum, I think the proposed solution is simple and straightforward. Thus, I think the proposed approach is easily reproducible. The particular strengths I see are the following:
1. Point-wise positive and negative pairs to balance both region- and scene-level contrasts.
2. Strong empirical results across many datasets compared to prior work.
3. The paper includes ablation experiments on several of the components

**Weaknesses:**

I have some concerns about the proposed method:
1. How does it identify the semantic-less and semantic-rich points? How does it calculate the 4th dimension of the points? What is the range of 4th dimension of the points?
2. Is T2 identity transform in Fig. 1? If not, how does it generate the images from the original images? Were any related augmentations applied to the images, when the lidar points were changed?

**Questions:**

See the weakness.

---

> ### Author Response · Authors · 2023-11-16
> **Response to Reviewer ksX5**
>
> We are glad that the reviewer found our method effective and comprehensive. We appreciate the opportunity to address the points you have raised.
>
> > How does it identify the semantic-less and semantic-rich points? How does it calculate the 4th dimension of the points? What is the range of the 4th dimension of the points?
>
> In our approach, the differentiation between semantic-rich and semantic-less points is achieved through a novel unsupervised technique termed **semantic pooling**. This process is designed to identify points potentially associated with objects of interest in an unsupervised manner, such as cars, cyclists, and pedestrians, categorizing them as *semantic-rich*. As our semantic pooling is based on the modified DBSCAN clustering algorithm with heuristics, these pooled points are extracted into different clusters based on distance metrics. The representation of these clusters can be seen in different colors in Figures 6, 7, and 8 in Appendix C. Points that are not selected by our semantic pooling, typically associated with less relevant features like buildings, ground, or grass, are marked as *semantic-less*.
>
> Regarding the 4th dimension of the points, it serves as a cluster identifier for the semantic-rich points. Each point is assigned an integer value in this dimension that corresponds to the cluster it belongs to, as determined by our semantic pooling operation. For semantic-less points, this value is set to -1, distinguishing them from their semantic-rich counterparts. It's important to note that the 4th dimension is used as metadata during our pretraining process and does not directly feed into the model learning.
>
> We have recognized the need for a more detailed explanation of these concepts and have included additional clarifications in the revised version of our paper.
>
> > Is $T_2$ identity transform in Fig. 1? If not, how does it generate the images from the original images?
>
> The second transformation $T_2$  does not necessarily need to be the identity transformation. We would like to clarify that Figure 1 is only for illustrative purposes. There are three transformations in the actual implementation of our framework.
>
> In the first stage of CALICO, namely point-region contrast (PRC), we employ two transformations, $T_1$ and $T_2$, to facilitate effective contrastive pretraining.
>
> In the second stage of CALICO, which is the region-aware distillation (RAD), we introduce another transformation on the xy-plane. This transformation, denoted as $T_3$, is similar to $T_1$ and $T_2$ and serves as a data augmentation method for the LiDAR modality. Concurrently, the view transformation module in the camera modality, LSS [a,b], incorporates $T_3$ when transforming the features from the perspective view to the bird’s eye view (BEV). This ensures that the feature maps from both LiDAR and camera modalities remain well-aligned in the distillation process.
>
> We hope our response addresses all the comments and that the reviewer will consider raising the rating accordingly. We are more than glad to answer any further questions.
>
> [a] Philion, Jonah, and Sanja Fidler. "Lift, splat, shoot: Encoding images from arbitrary camera rigs by implicitly unprojecting to 3d." Computer Vision–ECCV 2020: 16th European Conference, Glasgow, UK, August 23–28, 2020, Proceedings, Part XIV 16. Springer International Publishing, 2020.
>
> [b] BEVFusion Codebase https://github.com/mit-han-lab/bevfusion/blob/main/mmdet3d/models/vtransforms/base.py (disclaimer: this link contains a public codebase that does not leak any information from the authors.)

---

> > ### Comment · Reviewer_ksX5 · 2023-11-21
> >
> > Thanks for your rebuttal. It solved my concerns. I tend to raise my rating.

---

> > > ### Author Response · Authors · 2023-11-21
> > > **Thank you!**
> > >
> > > We are glad to hear that our response has addressed your concerns and we look forward to your updated rating!

---

### Official Review · Reviewer_fxyA · 2023-10-31

**Soundness:** 4 excellent
**Presentation:** 4 excellent
**Contribution:** 3 good
**Rating:** 8
**Confidence:** 3

**Summary:**

The paper proposes a self-supervised learning method of Image and point-cloud input that consists of point-region contrast (PRC) and region-aware distillation (RAD).
Differing from the previous works, PRC utilizes both point- and region-level contrastive learning on point cloud. RAD aligns the feature maps of Images and points.
The proposed method is evaluated and shows substantial performance improvement on 3D detection and BEV map segmentation tasks on nuScene and Waymo datasets.
Ablation studies and robustness tests are also thorough to demonstrate the effectiveness of the proposed method.
After the author discussion phase, I will adjust or fix my decision.

**Strengths:**

[Originality]
+ Differing from the previous works, the proposed method welly utilizes both point- and region-level contrastive learning on point cloud.

[Quality & Significance]
+ The proposed method is evaluated and shows substantial performance improvement on 3D detection and BEV map segmentation tasks on nuScene and Waymo datasets.
+ Ablation studies and robustness tests are also thorough to demonstrate the effectiveness of the proposed method.

**Weaknesses:**

[Quality]
- The proposed method adopts the BEVFusion architecture. Lidar backbone is PointPillars, and Image backbone is Swin-T.
Evaluation of different Lidar/Image backbone models could have a high impact.
- The necessity of negative samples of P_PLRC and P_RAPC is unclear. The ablation study (w/o negative samples in P_PLRC and P_RAPC) supports the necessity.

**Questions:**

Please see the weakness part.

---

> ### Author Response · Authors · 2023-11-16
> **Response to Reviewer fxyA [1/2]**
>
> We are glad that the reviewer found our method effective and comprehensive. We appreciate the opportunity to address the points you have raised.
>
> > The proposed method adopts the BEVFusion architecture. Lidar backbone is PointPillars, and Image backbone is Swin-T. Evaluation of different Lidar/Image backbone models could have a high impact.
>
> We adopted the BEVFusion codebase for its state-of-the-art performance and its versatility as a general sensor fusion framework at the time of our implementation. In Section 3.5 of our paper, we have conducted an ablation study using the VoxelNet backbone and Transfusion head to specifically demonstrate the effectiveness of our point-region contrast (PRC) approach in the LiDAR modality. The results, as illustrated in Figure 4, clearly show that our PRC method consistently outperforms other baseline approaches in terms of performance.
> Additionally, in response to your valuable suggestion, we have integrated ResNet as an alternative backbone for the camera modality (as configured in the BEVFusion codebase). Specifically, we use our PRC pretrained LiDAR backbone as the teacher model and leverage the 10% finetuning data setting in this experiment.
> This further step is taken to demonstrate the adaptability and generalizability of our framework across different backbone architectures.
>
> | ResNet as the Camera Backbone with 10% Finetuning Data  | mAP  | NDS  |
> |---------------------------------------------------------|------|------|
> | **PRC**+Rand. Init. (C)                                         | 47.9 | 52.0 |
> | SimIPU                                                  | 46.6 | 51.4 |
> | **PRC**+BEVDistill                                          | 48.5 | 52.5 |
> | **CALICO**                                                  | **49.0** | **53.1** |
>
> The results in the above table illustrate a consistent trend in performance between Swint-T and ResNet as the image backbone. More importantly, our CALICO using the ResNet backbone consistently achieves the best performance compared to other baseline methods.

---

> ### Author Response · Authors · 2023-11-16
> **Response to Reviewer fxyA [2/2]**
>
> > The necessity of negative samples of $P_{PLRC}$ and $P_{RAPC}$ is unclear.
>
> To address this, we first acknowledge the fundamental role of negative samples in contrastive learning. The absence of negative samples can lead to model collapse, as highlighted in several studies [a, b, c]. Additionally, methods focusing solely on positive pairs often implicitly benefit from statistics hidden negative samples during model training [d].
>
> Therefore, we hypothesize that you are specifically inquiring about the necessity and role of negative sample augmentation through semantic-less points in our framework. Our approach addresses a unique challenge in LiDAR data, notably the presence of multiple objects of interest within a single frame. For instance, a LiDAR frame capturing a traffic intersection may contain numerous cars and pedestrians. If we restrict our framework to only use pooled points for generating negative pairs, we risk encountering the **class collision** problem. This occurs when points from different clusters (extracted by our semantic pooling), although distinct, belong to the same object class (e.g., car). There will be performance degradation when treating points from the same class of object as negative pairs. Our negative sample augmentation is thus proposed to alleviate this problem.
>
> To empirically validate our approach and address the concerns raised, we conducted experiments specifically focusing on the impact and necessity of our negative sample augmentation strategy.
>
> | 10% Finetuning Data                               | mAP  | NDS  |
> |---------------------------------------------------|------|------|
> | PLRC                                              | 41.9 | 50.9 |
> | + semantic pooling                                | 42.8 | 51.5 |
> | + semantic pooling + negative sample augmentation | **43.3** | **51.9** |
>
> | 50% Finetuning Data                               | mAP  | NDS  |
> |---------------------------------------------------|------|------|
> | PLRC                                              | 52.5 | 60.0 |
> | + semantic pooling                                | 53.0 | 60.5 |
> | + semantic pooling + negative sample augmentation | **53.2** | **60.8** |
>
> These experimental results clearly demonstrate the effectiveness and necessity of our negative sample augmentation method in improving the overall model performance.
>
> We have included all the experimental results in the revised manuscript and hope our response addresses all the comments and that the reviewer will consider raising the rating accordingly. We are more than glad to answer any further questions.
>
> [a] He, Kaiming, et al. "Momentum contrast for unsupervised visual representation learning." Proceedings of the IEEE/CVF conference on computer vision and pattern recognition. 2020.
>
> [b] Chen, Ting, et al. "A simple framework for contrastive learning of visual representations." International conference on machine learning. PMLR, 2020.
>
> [c] Grill, Jean-Bastien, et al. "Bootstrap your own latent-a new approach to self-supervised learning." Advances in neural information processing systems 33 (2020): 21271-21284.
>
> [d] Li, Alexander C., Alexei A. Efros, and Deepak Pathak. "Understanding collapse in non-contrastive siamese representation learning." European Conference on Computer Vision. Cham: Springer Nature Switzerland, 2022.

---

> > ### Comment · Reviewer_fxyA · 2023-11-21
> >
> > Thanks for your rebuttal. My concerns are resolved!

---

> > > ### Author Response · Authors · 2023-11-21
> > > **Thank you!**
> > >
> > > We are glad to hear that our response has addressed your concerns. We would like to thank you again for the constructive feedback!

---

### Official Review · Reviewer_2HK8 · 2023-11-04

**Soundness:** 3 good
**Presentation:** 3 good
**Contribution:** 2 fair
**Rating:** 6
**Confidence:** 5

**Summary:**

In this paper, a method for LiDAR-Camera BEV fusion is explored via contrastive and self-supervised training. Using currently fashionable machinery - Lift Splat Shoot type camera encoder, Voxelnet LiDAR encoder, Transfusion decoder, etc. - for feature processing in the BEV literature, the paper adapts contrastive learning ideas for the problem. At a high level, delineations are made between between point level and region level contrast, with heuristics (e.g. clustering) being applied to extract more discriminative features.

Evaluations are presented to compare with other contrastive methods on the NuScenes and Waymo datasets. After contrastive (unsupervised/self-supervised) pretraining, the setup is fine tuned for object detection and segmentation tasks with varying amounts of training data to show efficacy of the pretraining step. Furthermore, they also investigate robustness to adversarial attacks (inserting fake objects at some distance from ego vehicle) and corruption of data (as might occur in bad weather, degraded sensors).

**Strengths:**

+ Contrastive training is generally less studied in the BEV perception literature. This work adds to the body of work present in the area.
+ The methods are generalizable, and can be applied to any setup.
+ Effectiveness is shown across modalities in camera, camera+lidar and lidar. This is convincing. I was particularly impressed with the saliency maps with and without pre-training.

**Weaknesses:**

- The paper is entirely empirical, and is as such a purely application based work. One may or may not take this as a weakness, of course.
- On the same lines as above, the paper is heavy on tables, but I feel that qualitative analysis of where the improvement comes from is light. Some analysis through carefully designed experiments that show improvement with and without various fittings would shed insight. It appears that semantic pooling (feature rich vs feature less regions) plays a part from the figure 5, but I would like more examples of failure cases.

**Questions:**

- Could the authors explain how adversarial robustness is relevant in this context?
- The clustering methods look rather empirical. More experiments on how they work in different cases would be useful.
- Calibration error analysis: I think it would help to learn if the system can demonstrate robustness against calibration error, a common occurrence in autonomous driving setups.
- Superfluous lines from possible prior submission (Appendix C, under 'Visualization')

"As we mentioned in the rebuttal,camera features trained from scratch are not salient to contribute to robustness improvement."

---

> ### Author Response · Authors · 2023-11-16
> **Response to Reviewer 2HK8 [1/2]**
>
> We are glad that the reviewer found our study valuable to this field and that our method is effective and generalizable. We appreciate the opportunity to address the points you have raised.
>
> > The paper is entirely empirical, and is as such a purely application based work. One may or may not take this as a weakness, of course.
>
> We agree with the reviewer's observation regarding the empirical nature of our study. While it is true that our paper focuses on application-driven research, we believe this approach is not only valid but also crucial in the context of pretraining methods. As the reviewer notes, the empirical focus should not inherently be seen as a weakness.
>
> In the field of pretraining, whether contrastive (e.g., MoCo [a]) or generative (e.g., MAE [b]), the practical application of methods often precedes and informs theoretical understanding. This is particularly true in rapidly advancing areas where empirical results can guide and refine theoretical models. Our work contributes to this tradition by providing valuable insights through real-world scenarios in autonomous driving perception.
>
> To comply with the suggestions from the reviewer, we have included a new discussion in Appendix D.1 of our revised manuscript on potential avenues for theoretical exploration in pretraining methods, acknowledging the current limitations while highlighting the importance of empirical contributions like ours in laying the groundwork for future theoretical advancements.
>
> >  The paper is heavy on tables, but I feel that qualitative analysis of where the improvement comes from is light.
>
> We appreciate the reviewer's insightful observation regarding the balance between quantitative and qualitative analysis in our paper. We agree that the abundance of tables could potentially overshadow the qualitative insights. The choice to emphasize tabular data was driven by our intent to provide a comprehensive and detailed comparison across different metrics.
>
> In response to the reviewer's suggestion, we have included a more focused qualitative analysis in Appendix B.1. This analysis presents key insights derived from our results. Notably, we observe that the benefits of pretraining diminish with the increase in fine-tuning data. This highlights the nuanced trade-off between pretraining and fine-tuning in model performance, which is as expected. Furthermore, when comparing our proposed PRC method (augmented with the RAPC component) to the PLRC approach, it becomes evident that our method demonstrates more significant improvements, especially when finetuning on 50% of the data. Moreover, our ablation study in Table 5 also showcases that the RAPC component in our design serves as a good regularization term to balance the performance of our framework among different levels of availabilities in finetuning data.
>
> We hope that this enhanced qualitative discussion complements the extensive quantitative data presented and provides a more holistic understanding of our research contributions.
>
> > Could the authors explain how adversarial robustness is relevant in this context?
>
> The relevance of adversarial robustness in our study is directly tied to the phenomenon of LiDAR spoofing attacks, as presented in [c]. These attacks involve the injection of malicious data points into LiDAR sensors, posing significant driving hazards. Our work included an evaluation of the resilience of models pretrained using our CALICO framework, as well as other baseline methods, against these black-box sensor attacks. To simulate the real-world implications of such attacks, we crafted scenarios with varying degrees of complexity, introducing 60, 100, and 200 spoofing points to create the illusion of false objects 10 meters in front of the ego vehicle, adhering to the protocols established in [c].
>
> Our results are promising. We observed that CALICO notably diminished the success rates of these attacks in comparison to models trained from scratch and LiDAR-only methods. This increased robustness can be attributed to CALICO's ability to achieve a **more balanced** pretraining process, integrating both LiDAR and camera modalities effectively. In contrast, models trained from scratch or using LiDAR-only pretraining methods exhibited a higher susceptibility to attacks, because we found their end-to-end models mainly or entirely rely on the LiDAR modality.
>
> In conclusion, while CALICO's primary aim is to enhance the efficiency and accuracy of 3D object detection models, it also plays a crucial role in fortifying these models against sophisticated adversarial attacks.

---

> ### Author Response · Authors · 2023-11-16
> **Response to Reviewer 2HK8 [2/2]**
>
> >  It appears that semantic pooling (feature rich vs feature less regions) plays a part from the figure 5, but I would like more examples of failure cases. The clustering methods look rather empirical. More experiments on how they work in different cases would be useful.
>
> In our exploration of the unsupervised nature of our semantic pooling operation in the CALICO framework, we encountered certain limitations. For instance, the semantic pooling sometimes erroneously clusters tree points as semantic-rich. Nonetheless, these instances did not significantly impact overall performance. The primary objective during the pretraining phase is to learn robust and prominent representations within the model’s backbone, a process that remains largely independent from downstream tasks. Misclassifying a small number of random objects as semantic-rich does not detrimentally affect the contrastive pretraining for the model backbone because the objective is to learn the **correspondence** between regions. Furthermore, the core concept of semantic pooling is pivotal, transcending the specifics of its implementation. Our current implementation utilized a basic approach to achieve region partitioning in LiDAR BEV space relying on independent LiDAR frames. Enhancements to semantic pooling could include integrating temporal data to refine point aggregation or to better aggregate points and identify moving points. However, such an approach necessitates temporally continuous data, introducing an additional layer of assumption and complexity. CALICO is designed to be versatile with respect to the choice of clustering algorithms, as long as the region partitioning is reasonable, as the core of CALICO is its following contrastive learning design based on the partitioned regions.
>
> We have added an ablation study in Appendix B to quantitatively demonstrate the effectiveness of our semantic pooling compared to the random partitioning leveraged in original PLRC. We have included visualizations and a discussion of this in our revised manuscript, as shown in Figure 7 and 8  in Appendix C. In particular, Figure 7 visualizes a LiDAR frame with some failure clusters and Figure 8 showcases that our semantic pooling could successfully cluster some small but interesting objects.
>
> > I think it would help to learn if the system can demonstrate robustness against calibration error, a common occurrence in autonomous driving setups.
>
> We are grateful for the reviewer's insightful suggestion to assess our system's resilience to calibration errors, a challenge in autonomous driving systems. Initially, it's worth clarifying that the primary objective of our framework is to enhance the efficiency and overall performance of self-driving perception tasks, with the robustness analysis serving as a supplementary aspect.
>
> Our current evaluation framework operates under the premise of accurate alignment between LiDAR and camera data. This alignment is pivotal for the efficacy of our pretraining methodology, particularly due to the necessity of data synchronization for successful knowledge distillation in RAD. In scenarios where we intentionally alter the calibration matrix between the two modalities, our method demonstrates similar performance to other baseline approaches.
>
> On one hand, we acknowledge that in real-world scenarios, misalignments due to calibration errors can occur, potentially impacting the performance of systems like ours. Although our current study did not specifically focus on this aspect, we recognize its importance in the broader context of autonomous driving. On the other hand, evaluating calibration errors is challenging, as misalignments result in differing localization ground truths for both modalities (i.e., the ground truth bounding boxes also do not align with each other). Therefore, determining a method and metric to quantify the evaluation is controversial.
>
> In response to the reviewer's valuable feedback, we have incorporated the above discussion in our revised manuscript about the potential implications of calibration errors on systems pre-trained using CALICO. This addition not only acknowledges the issue but also underlines it as a vital area for future research, aiming to further refine the robustness and reliability of autonomous driving technologies.
>
> We thank the reviewer for pointing out the superfluous lines in our manuscript and we have removed them in our revised manuscript.
>
> We hope our response addresses all the comments and that the reviewer will consider raising the rating accordingly. We are more than glad to answer any further questions.

---

> ### Author Response · Authors · 2023-11-16
> **References**
>
> [a] He, Kaiming, et al. "Momentum contrast for unsupervised visual representation learning." Proceedings of the IEEE/CVF conference on computer vision and pattern recognition. 2020.
>
> [b] He, Kaiming, et al. "Masked autoencoders are scalable vision learners." Proceedings of the IEEE/CVF conference on computer vision and pattern recognition. 2022.
>
> [c] Sun, Jiachen, et al. "Towards robust {LiDAR-based} perception in autonomous driving: General black-box adversarial sensor attack and countermeasures." 29th USENIX Security Symposium (USENIX Security 20). 2020.

---

> ### Comment · Reviewer_2HK8 · 2023-11-22
> **Further clarifications**
>
> Thank you for your detailed and thoughtful response. I still feel that the work is very empirical and does not shed theoretical insight (despite the additional detail provided in the appendix). Nonetheless, I really enjoyed reading the paper and can see that distillation + sensible ideas such as clustering could be very useful in a practical problem such as sensor fusion.
>
> The ablations in appendix B show that the distillation idea works for different backbones. I would have hoped for more ablations on methods but am fairly convinced with the response.
>
> Perhaps I am not reading this correctly, but I notice that the paper does not beat (nor even match) BEVFusion's results - the paper reports an mAP of 60.1 (table 1) for Lidar+Camera, whereas in BEVFusion, they get a number of 75 or so (table 1 of BEVFusion [1]). As the paper uses the BEVFusion architecture, one might hope that it can match BEVFusion's results. Could the authors explain this? Can the authors also report distance based results - how does detection perform in near and far distances, and per-class metrics (car, pedestrian, etc.)? Can the authors shed light on these aspects?
>
>
> [1] BEVFusion: https://arxiv.org/pdf/2205.13542.pdf

---

> > ### Author Response · Authors · 2023-11-23
> > **Response to Follow-up Questions [1/2]**
> >
> > We thank the reviewer for their follow-up comments and we further respond to each point in the following:
> >
> > >  I still feel that the work is very empirical and does not shed theoretical insight.
> >
> > We appreciate the reviewer's recognition of the practical contributions of our work, particularly regarding the application of distillation techniques and clustering strategies. We acknowledge the reviewer's perspective on the empirical nature of our study. Indeed, our work primarily focuses on the practical implementation and efficacy of these methods in real-world scenarios, which inevitably leans more towards empirical analysis.
> >
> > We concur that providing deeper theoretical insights could enrich the understanding of why and how these methods work effectively. This aspect, we believe, is a **common limitation** in the field of pretraining methods, where the emphasis often lies in demonstrating empirical effectiveness rather than delving into the underlying theoretical foundations.
> >
> > Moving forward, we see this as an opportunity for future research. Our findings lay the groundwork for subsequent studies that could bridge this gap by integrating theoretical analysis with empirical results.
> >
> > > Perhaps I am not reading this correctly, but I notice that the paper does not beat (nor even match) BEVFusion's results - the paper reports an mAP of 60.1 (table 1) for Lidar+Camera, whereas in BEVFusion, they get a number of 75 or so (table 1 of BEVFusion [1]). As the paper uses the BEVFusion architecture, one might hope that it can match BEVFusion's results. Could the authors explain this?
> >
> > We appreciate the reviewer's thorough analysis and observation. Allow us to clarify the apparent discrepancy in performance metrics compared to the original BEVFusion paper. Our approach involves using a basic BEVFusion architecture using **PointPillars** as the LiDAR backbone **without** test-time adaptation (TTA) and our evaluations are conducted on the validation set. More crucially, as the relevant benchmark in Table 1 of BEVFusion [1], which reports an mAP of 68.5 and an NDS of 71.4 (**VoxelNet** as the LiDAR backbone without TTA on the validation set), is based on training using **100%** of the labeled training set. We use PointPillars as it is more efficient than VoxelNet as we have many more configurations to pretrain, finetune, and evaluate than the original BEVFusion paper. We have also included a small ablation study in Section 3.5 to demonstrate that our CALICO can also work with VoxelNet and the results are better than PointPillars (the first paragraph in Section 3.5).
> >
> > In contrast, as detailed in our Tables 1 and 2, our methodology entails finetuning our pretrained backbone on a much more limited portion of the training set, ranging from **5% to 50%**. This strategy of employing restricted labeled data for finetuning is a standard practice in research focusing on pretraining methods in self-driving perception, as substantiated by references [2,3,4]. Given this significant reduction in training data, a modest decrease in performance metrics compared to the full-data scenario is to be anticipated and is in line with current understanding in the field.
> >
> > To directly address the reviewer's concern, we have also conducted comparative analyses where models are trained from scratch versus those finetuned using our CALICO-pretrained backbones, specifically on the 5% to 50% segment of the training dataset without TTA and with PointPillars as the LiDAR backbone.
> >
> > | Labeled Data fro Training/Finetuning | Method                      | NDS      | mAP      |
> > |--------------------------------------|-----------------------------|----------|----------|
> > | 5%                                   | BEVFusion (Rand. Init. L+C) | 42.8     | 35.1     |
> > | 5%                                   | **BEVFusion + CALICO**      | **47.9** | **41.7** |
> > | 10%                                  | BEVFusion (Rand. Init. L+C) | 51.0     | 46.2     |
> > | 10%                                  | **BEVFusion + CALICO**      | **53.9** | **50.0** |
> > | 20%                                  | BEVFusion (Rand. Init. L+C) | 58.0     | 51.3     |
> > | 20%                                  | **BEVFusion + CALICO**      | **59.5** | **54.8** |
> > | 50%                                  | BEVFusion (Rand. Init. L+C) | 61.7     | 57.5     |
> > | 50%                                  | **BEVFusion + CALICO**      | **62.7** | **60.1** |
> >
> > This comparison provides a more nuanced understanding of our model's efficacy within the context of limited data availability.
> >
> > [1] BEVFusion: https://arxiv.org/pdf/2205.13542.pdf
> >
> > [2] Exploring Geometry-aware Contrast and Clustering Harmonization for Self-supervised 3D Object Detection
> >
> > [3] PointContrast: Unsupervised Pre-training for 3D Point Cloud Understanding
> >
> > [4] ProposalContrast: Unsupervised Pre-training for LiDAR-based 3D Object Detection

---

> > > ### Author Response · Authors · 2023-11-23
> > > **Response to Follow-up Questions [2/2]**
> > >
> > > > Also, I would very much like to see numbers with competitive methods - i.e. to show that distillation works independently of the method used.
> > >
> > > Table 7 in Appendix B has presented an ablation study on our region-aware distillation method compared to other distillation methods and camera-only object detectors. BEVDistill should be the state-of-the-art distillation framework, to the best of our knowledge. The results in Table 7 fairly demonstrate that our RAD outperformed the other baseline methods. We are open to any baseline suggestions for us to add into our manuscript. However, as the discussion stage is approaching the end, we have to include additional experiments in our camera-ready revision if accepted.
> > >
> > > > Can the authors also report distance based results - how does detection perform in near and far distances, and per-class metrics (car, pedestrian, etc.)? Can the authors shed light on these aspects?
> > >
> > > We thank the reviewer for this valuable suggestion. We summarize our results in the following tables. Specifically, we breakdown the results into 10 classes and follow BEVFusion to breakdown the distance into three levels.
> > >
> > > | Labeled Data fro Training/Finetuning | Method (NDS)                  | Car  | Truck | Bus  | Trailer | Construction Vehicle | Pedestrian | Motorcycle | Bicycle | Traffic Cone | Barrier |
> > > |--------------------------------------|-----------------------------|------|-------|------|---------|----------------------|------------|------------|---------|--------------|---------|
> > > | 10%                                  | BEVFusion (Rand. Init. L+C) | 79.2 | 31.8  | 45.7 | 39.8    | 11.2                 | 73.5       | 37.9       | 7.9     | 57.6         | 47.9    |
> > > | 10%                                  | **BEVFusion + CALICO**      | 80.0 | 33.6  | 46.8 | 41.8    | 15.7                 | 76.2       | 40.7       | 10.1    | 58.8         | 48.8    |
> > > | 50%                                  | BEVFusion (Rand. Init. L+C) | 81.3 | 48.3  | 54.5 | 41.5    | 20.6                 | 79.3       | 50.9       | 20.5    | 69.5         | 63.3    |
> > > | 50%                                  | **BEVFusion + CALICO**      | 81.3 | 48.5  | 54.9 | 42.9    | 24.5                 | 79.6       | 51.5       | 22.3    | 70.9         | 64.5    |
> > >
> > >
> > > | Labeled Data fro Training/Finetuning | Method (mAP)                     | 0-20m | 20-30m | >30m |
> > > |--------------------------------------|-----------------------------|-------|--------|------|
> > > | 10%                                  | BEVFusion (Rand. Init. L+C) | 56.9  | 42.5   | 25.4 |
> > > | 10%                                  | **BEVFusion + CALICO**      | 59.7  | 45.1   | 31.2 |
> > > | 50%                                  | BEVFusion (Rand. Init. L+C) | 66.4  | 52.3   | 37.8 |
> > > | 50%                                  | **BEVFusion + CALICO**      | 67.8  | 55.3   | 41.0 |
> > >
> > >
> > > The tables provided offer a detailed breakdown of our findings. Notably, when training or finetuning with a limited amount of labeled data, our approach shows significant improvements, particularly in detecting smaller and more distant objects. As the volume of labeled data increases, we observe a more balanced enhancement across various classes and distances.

---

> > > ### Comment · Reviewer_2HK8 · 2023-11-23
> > >
> > > Thank you for the clarifications. It makes sense that the numbers are lower as the full train set wasn't used.

---

### Author Response · Authors · 2023-11-16
**General Response to All Reviewers**

Thank you so much for your constructive and invaluable feedback, which we have taken into consideration and incorporated into our revised manuscript. We appreciate the recognition by all reviewers of our method's effectiveness and generalizability. In addition to specifically addressing each reviewer's queries, we have made the following updates to our manuscript, reflecting the constructive suggestions provided:

1. In response to Reviewer 2HK8's input, we have introduced Appendix B.1, offering an expanded qualitative analysis.

2. Further, to address Reviewer 2HK8's request, we have enhanced our explanation of the semantic pooling operation, complemented by additional visual examples (e.g., special and failure cases) in Appendix C.

3. Recognizing the need for a discussion on the limitation in theoretical analysis and calibration robustness of current pretraining designs, we have included Appendix D.1, which delves into this aspect, based on the feedback from Reviewer 2HK8.

4. Based on Reviewers 2HK8 and fxyA's suggestions, we have incorporated two new ablation studies in Appendix B. These studies examine the impacts of different camera backbones, semantic pooling, and negative sample augmentation on our design.

5. We have refined the introduction to semantic pooling in Section 2.2, ensuring a clearer explanation of the notation associated with the fourth dimension of each point, as suggested by Reviewer ksX5.

All modifications in the revised manuscript are highlighted in blue for ease of identification. We trust that these updates, alongside our detailed responses, have comprehensively addressed the reviewers' comments. We remain open and available to clarify any further queries or follow-up questions.

---

> ### Comment · Area_Chair_cLYn · 2023-11-19
> **Author-Reviewer Discussions (Fri, Nov 10 – Wed, Nov 22)**
>
> Dear reviewers,
> Could you please read the authors' responses and give your feed back? The period of Author-Reviewer Discussions is Fri, Nov 10 – Wed, Nov 22.
> Many thanks,
> AC

---

### Meta-Review · Area_Chair_cLYn · 2023-12-05

**Metareview:**

This paper proposes CALICO, a novel framework that applies contrastive objectives to both LiDAR and camera backbones.  CALICO contains point-region contrast (PRC) and region-aware distillation (RAD). Extensive evaluation results confirm the efficacy of the proposed methods.

This paper receives two “marginally above the acceptance threshold” ratings and one “accept, good paper” rating.

Reviewer 2HK8 gives “marginally above the acceptance threshold”. Reviewer 2HK8 thinks this work adds to the body of work present in the BEV perception literature. Reviewer 2HK8 thinks the proposed methods are generalizable. Reviewer 2HK8 is impressed with the experimental results. But also, Reviewer 2HK8 wants to see more qualitative analysis and more examples of failure cases. The authors gave their rebuttals. After reading them,  Reviewer 2HK8 has no further questions.

Reviewer fxyA gives “accept, good paper”. Reviewer fxyA thinks the proposed methods are novel. Reviewer fxyA is satisfied with the extensive experimental results. But also, Reviewer fxyA wants to see evaluation of different Lidar/Image backbone models. And, Reviewer fxyA wants to see negative samples of P_PLRC and P_RAPC and more ablation studies. The authors gave their rebuttals. After reading them,  Reviewer  fxyA has no further questions.

Reviewer ksX5 gives “marginally above the acceptance threshold”. Reviewer ksX5 thinks the proposed approach is easily reproducible. Reviewer ksX5 thinks the experiments are enough. But also,  Reviewer ksX5 raises some concerns. The authors gave their rebuttals. After reading them, Reviewer ksX5 has no further questions.

Therefore, based on the reviewers’ comments, the paper can be accepted by ICLR.

**Justification For Why Not Higher Score:**

This paper lacks some qualitative analysis, some examples of failure cases and some ablation studies mentioned by reviewers.

**Justification For Why Not Lower Score:**

The idea of this paper is novel. The proposed methods are generalizable. The experimental results confirms the efficacy of the proposed methods.

---

### Decision · Program_Chairs · 2024-01-16

Accept (poster)